# QTL Analysis for Rice Quality-Related Traits and Fine Mapping of *qWCR3*

**DOI:** 10.3390/ijms25084389

**Published:** 2024-04-16

**Authors:** Jun Liu, Hao Zhang, Yingying Wang, Enyu Liu, Huan Shi, Guanjun Gao, Qinglu Zhang, Guangming Lou, Gonghao Jiang, Yuqing He

**Affiliations:** 1National Key Laboratory of Crop Genetic Improvement and National Center of Plant Gene Research, Hubei Hongshan Laboratory, Huazhong Agricultural University, Wuhan 430070, China; 15528018871@163.com (J.L.); 15071031326@163.com (H.Z.); ley121108260823@163.com (E.L.); qingluzhang@mail.hzau.edu.cn (Q.Z.); louguangming@mail.hzau.edu.cn (G.L.); 2Institute of Crop Germplasm Resources, Guizhou Academy of Agriculture Science, Guiyang 550006, China; 3College of Life Science, Heilongjiang University, Harbin 150080, China

**Keywords:** rice quality, genetic linkage analysis, fine mapping, *qWCR3*, chalkiness

## Abstract

The quality of rice, evaluated using multiple quality-related traits, is the main determinant of its market competitiveness. In this study, two *japonica* rice varieties with significant differences in quality-related traits were used as parents to construct two populations, BC_3_F_2_ and BC_3_F_2:3_, with Kongyu131 (KY131) as the recurrent parent. A genetic linkage map was constructed using the BC_3_F_2_ population based on 151 pairs of SSR/InDel polymorphic markers selected between the parents. Grain-shape-related traits (grain length GL, grain width GW, and length-to-width ratio LWR), chalkiness-related traits (white-core rate WCR, white-belly rate WBR, white-back rate BR, and chalkiness rate CR), and amylose content (AC) were investigated in the two populations in 2017 and 2018. Except for BR and CR, the traits showed similar characteristics with a normal distribution in both populations. Genetic linkage analysis was conducted for these quality-related traits, and a total of 37 QTLs were detected in the two populations. Further validation was performed on the newly identified QTLs with larger effects, and three grain shape QTLs and four chalkiness QTLs were successfully validated in different environments. One repeatedly validated QTL, *qWCR3*, was selected for fine mapping and was successfully narrowed down to a 100 kb region in which only two genes, *LOC_0s03g45210* and *LOC_0s03g45320*, exhibited sequence variations between the parents. Furthermore, the variation of *LOC_Os03g45210* leads to a frameshift mutation and premature protein termination. The results of this study provide a theoretical basis for positional cloning of the *qWCR3* gene, thus offering new genetic resources for rice quality improvement.

## 1. Introduction

As one of the main food crops, rice (*Oryza sativa* L.) feeds billions of people worldwide [1]. With increasing living standards, the demand of consumers has shifted the goal of rice breeding toward the improvement of rice grain quality [2]. Rice quality is a complex trait that includes milling quality, appearance quality, nutrition quality, eating and cooking quality (ECQ), and hygiene quality [3]. Among these, appearance quality (which is mainly determined by grain shape and chalkiness [4]) and ECQ (which is mainly determined by the amylose content) are the primary factors affecting the market value of rice [3].

Grain shape, which affects rice yield and quality simultaneously, is determined according to three traits: grain length, grain width, and grain thickness. Although a great number of major genes regulating grain shape have been cloned in recent years, there are still many minor genes to be discovered [5].

Chalkiness can be divided into white-belly, white-core, and white-back based on its distribution on the rice grain [6]. Chalkiness is an undesirable trait for consumers that negative affects rice grain appearance quality, milling quality, and ECQ [7]. The formation of chalkiness is regulated by both genetic factors and environmental factors [8]. Over past few decades, many chalkiness QTLs (quantitative trait loci) have been identified [9], but only a major chalkiness gene *Chalk5* and a major white-core gene *WCR1* have been cloned [10,11]. *Chalk5* encodes a vacuolar H^+^-translocating pyrophosphatase (V-ppase) which disrupts the pH homeostasis of the developing endomembrane transport system, thereby affecting the biogenesis of protein bodies and forming air gaps between storage component in the endosperm, ultimately leading to chalkiness [10]. *WCR1* encodes an F-box protein. OsDOF17 binds to the *WCR1* promoter to increase its transcription level to promote the transcription of *MT2b* and inhibit 26S proteasome-mediated MT2b degradation, promoting the elimination of excess ROS, maintaining redox homeostasis, and delaying programmed cell death (PCD) in the endosperm, ultimately leading to a decrease in the white-core rate [11]. Thus, discovering more chalkiness genes is conducive to deepening our understanding of rice chalkiness.

Eating and cooking quality is reflected in multiple indices, including amylose content (AC), gel consistency (GC), gelatinization temperature (GT), and aroma and taste value [3]. AC is mainly regulated by a gene specifically expressed by the endosperm, *Wx*/*GBSSI,* which fluctuates between 0% to 30% according to the different allele of *Wx* [12]. However, there are also many minor genes that affect AC, such as *Osbzip58*, *OsBP5*, and *Du1* [13,14,15].

In this study, two back-cross populations derived from two rice cultivars (Cypress and KY131) were developed and used for QTL mapping of the rice grain shape, chalkiness, and AC. As a result, 18 grain shape QTLs, 18 chalkiness QTLs, and 1 AC QTL were detected. The genetic effects of three grain shape QTLs—*qGL7.2*, *qLWR7.2,* and *qGW3*—and three chalkiness QTLs—*qBR2*, *qWCR3*, and *qWCR11*—were then confirmed in a random population. Using the progeny test method, *qWCR3* was fine-mapped to a 100 kb region of chromosome 3.

## 2. Results

### 2.1. Phenotype Variation in BC_3_F_2_ and BC_3_F_2:3_ Populations

The grain length (GL), grain width (GW), length-to-width ratio (LWR), and amylose content (AC) had consistently higher values in Cypress than in KY131, while an opposite trend was observed for white-core rate (WCR), white-belly rate (WBR), white-back rate (BR), and chalkiness rate (CR) in both the BC_3_F_2_ (Hainan 2017) and BC_3_F_2:3_ populations (Wuhan 2018; Table 1). The GL, GW, and LWR in BC_3_F_2_ and BC_3_F_2:3_ populations showed a normal distribution (Figure 1A–F), as well as WBR, WCR, and AC in the BC_3_F_2:3_ population (Figure 1H,I,K); however, BR and CR showed skewed distributions in the BC_3_F_2:3_ population (Figure 1G,J).

Correlation analysis was conducted for all traits in the BC_3_F_2:3_ population. Among grain-shape-related traits, an extremely significant positive correlation was detected between GL and LWR, while an extremely significant negative correlation was found between GW and LWR. Among the chalkiness-related traits, BR was positively correlated to CR with high significance, and there were also significant positive correlations among WCR, WBR, and CR (Table 2). Moreover, GL was significantly positively correlated to WBR, while GW was negatively correlated to WBR and WCR with high significance.

### 2.2. Construction of Genetic Linkage Map

After screening nearly 1000 markers, 151 SSR (simple sequence repeat) and InDel (insertion and deletion) polymorphic makers between KY131 and Cypress were selected for genetic linkage map construction (Appendix A). As a result, the genetic map covered a 948.6 cM genome region with an average interval size of 6.3 cM (Figure 2).

### 2.3. 37 QTLs Were Detected in BC_3_F_2_ and BC_3_F_2:3_ Populations

Using the genetic linkage analysis method, a total of 37 QTLs were detected on 11 chromosomes (excluding chromosome 10) in the BC_3_F_2_ and BC_3_F_2:3_ populations (Table 3).

A total of 18 grain-shape-related QTLs were found, of which there were seven for GL (*qGL3*, *qGL4*, *qGL6*, *qGL7.1*, *qGL7.2*, *qGL7.3*, and *qGL8*); the phenotypic variance explained by each QTL for GL ranged from 4.55% to 20.60%, and *qGL7.1* was detected repeatedly both years. There were six QTLs for GW (*qGW1*, *qGW3*, *qGW5.1*, *qGW5.2*, *qGW7.1,* and *qGW7.2*), with *qGW3* and *qGW7.2* located in the same intervals as *qGL3* and *qGL7.2*, respectively. We also detected five QTLs for LWR (*qLWR3*, *qLWR5*, *qLWR7.1*, *qLWR7.2*, and *qLWR9*), of which *qLWR5* and *qLWR7.1* were detected repeatedly both years, and *qLWR7.1* was located in the same interval as *qGW7.1*.

A total of 18 QTLs for chalkiness were found in BC_3_F_2:3_ population, including four QTLs for BR (*qBR1*, *qBR2*, *qBR9,* and *qBR11*), four QTLs for WBR (*qWBR1.1*, *qWBR1.2*, *qWBR5,* and *qWBR6*), five QTLs for WCR (*qWCR3*, *qWCR5.1*, *qWCR5.2*, *qWCR11,* and *qWCR12*), and five QTLs for CR (*qCR1.1*, *qCR1.2*, *qCR2*, *qCR9,* and *qCR11*). Of these, *qBR2* and *qBR11* were located in common intervals with *qCR2* and *qWCR11*, respectively.

For AC, only one QTL (*qAC4*) was detected in the BC_3_F_2:3_ population, explaining 4.01% of the phenotype variation.

### 2.4. Genetic Effect Validation of Seven QTLs

Based on the results of QTL mapping, a number of QTLs with large effects were selected for genetic effect validation. The seeds of BC_3_F_2:3_ plants with heterozygous genotype in each QTL interval were selected to construct near-isogenic lines for genetic effect validation. As shown in Table 4, three grain-shape-related QTLs (*qGW3*, *qGL7.2*, and *qLWR7.2*) and four chalkiness-related QTLs (*qWBR1.2*, *qBR2*, *qWCR3,* and *qWCR11*) were chosen for genetic effect validation. In particular, *qWBR1.2*, *qGL7.2*, and *qGW3* were validated in one environment, while *qLWR7.2*, *qBR2*, *qWCR3,* and *qWCR11* were validated in two environments.

### 2.5. qWCR3 Was Fine-Mapped to a 100 kb Region

We selected a stable QTL, *qWCR3*, for further research. For the fine mapping of *qWCR3*, a BC_3_F_4_ population consisting of 200 individuals was developed, and eight recombinants between D3-6 and C3.26.6 were identified. All recombinants were used for progeny testing of *qWCR3*, and 10 InDel makers were developed to genotype these recombinants according to the sequence variation of the two parents (Figure 3A). The progeny test involves comparing the phenotypes of two homozygous genotype plants in the inbred line of each recombinant and, if there is a significant difference, the candidate gene is considered to be located in the heterozygous fragment of the recombinant; otherwise, it is considered to be in the homozygous fragment. Progeny test results are shown in Figure 3B, from which it can be seen that significant WCR differences were observed between homozygous plants of lines 5–1 and 5–2, while no significant WCR differences were detected in other recombinant lines, suggesting that the candidate gene should be located in the 100 kb region between makers 25.49 and 25.59, and that *qWCR3* co-segregated with maker 25.52.

### 2.6. LOC_Os03g45210 Could Be the Candidate Genes of qWCR3

The target 100 kb region of *qWCR3* contains 16 annotated genes (Table 5), according to the annotation information of the japonica variety Nipponbare from the RGAP website (Rice Genome Annotation Project; http://rice.uga.edu/; accessed on 8 April 2024). Of those, only two genes—*LOC_Os03g45210* and *LOC_Os03g45320*—exhibited sequence variation between Cypress and KY131. *LOC_Os03g45320* has a T insertion variation (Cypress-TTTTTTTTTT, KY131-TTTTTTTTTTT) on the intron between two parents, although it leads to no mutation of the protein. *LOC_Os03g45210* has an 11 bp InDel variation (CYPRESS-T, KY131-CATACAGGTTAT) on the second exon between two parents, leading to frameshift mutation and premature protein termination of *LOC_Os03g45210*. In addition, out of the 16 genes, only *LOC_Os03g45210* is specifically expressed in seed and endosperm, according to the prediction of the RGAP website (http://rice.uga.edu/; accessed on 8 April 2024). Therefore, we preliminary identified *LOC_Os03g45210* as the candidate gene for *qWCR3*, and the function of this gene needs to be further verified.

## 3. Discussion

Rice grain quality reflects the demands of consumers, sellers, and producers, consists of complex traits, and is one of the most important goals of rice breeding. Mapping-based cloning can help us identify QTLs (quantitative trait loci) that regulate rice grain quality traits, thus providing a theoretical basis for rice breeding. In this study, we identified 37 QTLs for grain shape, chalkiness, and amylose content in back-cross populations between Cypress and KY131.

### 3.1. Cloned Genes in the QTL Mapping Intervals

Among these QTLs, *qGL3* and *qGW3* were located in the same interval, which contains two cloned genes: *PGL1* and *BG1* [16,17]. *PGL1* encodes a bHLH protein which positively regulates grain length through regulation of cell length in the lemma [16]. *BG1* encodes a membrane localization protein, which plays a role in regulating the transport and distribution of auxin, thus positively regulating grain size [17]. The interval of *qGL6* contains a cloned gene, *BU1*, which regulates BR signaling pathways through *OsBRI1* and *RGA1* and controls the degree of curvature of rice leaf nodes [18]. The interval of *qGL8* contains a cloned gene, *GAD1,* which encodes a small secretory signal peptide leading to shorter grain length [19]. The interval of *qGW5.1* contains a cloned gene, *GS5,* which encodes a serine carboxypeptidase and positively regulates grain width [20]. *qGW5.2* and *qLWR5* are located in the same interval which contains two genes, *GSK2* and *OSCYP51G3,* that can regulate grain shape [21,22]. The intervals of *qGW5.1*, *qGW5.2,* and *qLWR5* share partial overlaps, with the overlapping area containing a grain width-controlling gene, *GW5*. A deletion mutation in *GW5* led to down-regulation of its expression level and to an increase in grain width. In addition, the GW5 protein can inhibit the activity of GSK2, thereby regulating the positive expression of genes in the downstream BR signaling pathway and positively regulating grain width [23]. The *qLWR3* interval contains a cloned gene, *GS3*. Through competitively binding with the rice G protein β subunit *RGB1*, GS3 inhibits the signal pathway of the G protein, thus regulating grain length [24,25]. The overlapping regions of *qGL7.2*, *qGW7.2*, *qLWR7.1,* and *qLWR7.2* contain five cloned genes, *GLW7*, *GL7*, *OsBZR1*, *BG2,* and *SRS1*/*DEP2*, which are related to grain shape [26,27,28,29,30]. These QTLs containing cloned genes may be used in rice breeding for the improvement of rice quality.

### 3.2. Four Chalkiness QTLs Were Newly Found

The formation of rice chalkiness is regulated by complex genetic networks and environments. Over the past few decades, researchers have detected a large number of quantitative trait loci controlling rice chalkiness [9]; however, few QTLs/genes have been fine-mapped or cloned. In this study, we detected 18 QTLs for chalkiness and verified four QTLs (*qWBR1.2*, *qBR2*, *qWCR3*, and *qWCR11*) repeatedly. The Cypress allele at *qWCR3* effectively increased the white-core rate (Table 3 and Table 4), while the Cypress alleles at *qWBR1.2*, *qBR2*, and *qWCR11* were found to effectively decrease the white-belly rate, white-back rate, and white-core rate, respectively. These results could serve as a basis for further fine mapping of the four QTLs and could be used in breeding programs to reduce chalkiness.

### 3.3. LOC_Os03g45210 Could Be a New Gene for Rice Chalkiness

*qWCR3* was fine-mapped to a 100 kb region, where only *LOC_Os03g45210* has an 11 bp InDel variation on the second exon between the two parents, leading to frameshift mutation and premature protein termination. *LOC_Os03g45210* encodes a protein belonging to the plant cysteine oxidase (PCOs) family, which act as O_2_-sensing enzymes in plants. The PCOs family in Arabidopsis controls the stability of ERF-VIIs through the N/Arg degradation pathway, first through oxidization of the N-terminal cysteine of ERF-VIIs proteins, then by carrying out Argyl-tRNA-protein transferase 1 (ATE1)-mediated arginine installation to control the stability of ERF-VIIs, thus regulating oxygen homeostasis [31]. When plants are submerged, the oxygen concentration level decreases and the activity level of PCOs decreases accordingly, leading to enhanced stability of ERF VIIs and accumulation of ethylene, allowing plants to adapt to the transcription of genes related to hypoxic conditions [32]. At present, the cloned white-core rate gene—*WCR1*—is known to be related to oxygen homeostasis in rice endosperm cells [11]. Taken together, the candidate gene of *qWCR3* may function as an O_2_ sensor in rice, controlling the formation of chalkiness through regulation of oxygen homeostasis in the endosperm.

## 4. Materials and Methods

### 4.1. Plant Materials and Field Experiment

A BC_3_F_1_ population consisting of 313 lines was derived from a cross between a temperate japonica cultivar KY131 (the recurrent parent) and a tropical japonica cultivar Cypress (the donor parent). The BC_3_F_2_ population consisted of 286 lines, with each line planted in one row including 12 plants. The BC_3_F_2_ population and their parents were planted in the experimental field of Huazhong Agricultural University in Lingshui city, Hainan province, in 2017. When the seeds of the BC_3_F_2_ plant matured, we mix-harvested the seeds of the BC_3_F_2_ lines to produce the BC_3_F_2:3_ lines, ensuring that the seeds of each of the BC_3_F_2_ lines corresponded strictly to a line of the BC_3_F_2:3_ population, i.e., each BC_3_F_2:3_ line originated from a mixed seed of a specific BC_3_F_2_ line. All the BC_3_F_2:3_ lines and their parents were planted in the experimental field of Huazhong Agricultural University in Wuhan city, Hubei province, in 2018. The phenotypes of rice quality-related traits in the BC_3_F_2_ and BC_3_F_2:3_ populations were collected for QTL mapping analysis.

In 2019, 2020, 2021, and 2022, the BC_3_F_2_ and BC_3_F_2:3_ lines with double heterozygous genotypes in the target QTL interval were planted for evaluation of genetic effects in the experimental field of Huazhong Agricultural University in Wuhan and Hainan. For the fine mapping of *qWCR3*, eight recombinants between D3-6 and C3.26.6 were identified from a BC_3_F_4_ population derived from a BC_3_F_3_ plant with a double heterozygous *qWCR3* segment, where all recombinants were planted in 8 rows with 12 plants per row.

The 30-day-old progeny seedlings of these populations were transplanted into eight rows with 12 seedlings each, having 16.5 cm spacing in single-row plots in the field, while the rows were 26.4 cm apart. Field management followed local practices. Ten plants from the middle of each row were harvested individually for the measurement of traits.

### 4.2. Phenotyping and Statistical Analysis

Before measurement, harvested grains were air-dried at room temperature for at least 3 months to maintain a consistent water balance. The grain length (GL), grain width (GW), and grain length-to-width ratio (LWR) were measured after threshing using a BenQ scanner and analyzed with SmartGrain software [33]. Grain chalkiness, including white-belly (WBR), white-core (WCR), and white-back (BR) phenotypes, was assessed visually. The percentage of chalky grains in the total number of dehulled grains was used for measurement of the grain chalkiness rate. The chalkiness rate (CR) was calculated as the sum of the three types of chalkiness. Flour ground from milled grain was used to measure the amylose content using the iodine staining method [34].

The grain shape, chalkiness, and amylose content of two different homozygous genotypes in the same QTL were compared through a *t*-test. If the *p*-value was less than 0.05, the candidate gene was considered to be located in the heterozygous fragment, otherwise it was considered to be located in the homozygous fragment.

### 4.3. Genotyping and QTL Analysis

Leaves from the parent varieties KY131 and Cypress were used to extract genome DNA using the CTAB method [35], and the population’s leaves were used to extract genome DNA using the TPS method. The parent genomes were sequenced and assembled according to the rice reference genome (Rice Genome Annotation Project; http://rice.uga.edu/; accessed on 8 April 2024) [36]. According to the sequencing data of the parents, we developed 151 polymorphic simple sequence repeat (SSR) and InDel makers distributed over all 12 chromosomes. The genome DNA was amplified through PCR, following a program of 94 °C for 5 min, then 32–34 cycles of 30 s at 94 °C, 30 s at 55 °C, 30 s at 72 °C, and, finally, 5 min at 72 °C. The PCR products were identified through 4% polyacrylamide gel electrophoresis (PAGE). Through the use of two flanking markers for each QTL, the corresponding genetic effect populations were genotyped.

### 4.4. Progeny Testing Analysis

Progeny testing was conducted in the progeny segregation population of each recombinant. In the progeny line of each recombinant, a student’s *t* test was performed to compare the grain size differences between plants with a homozygous KY131 allele and a homozygous Cypress allele. If the *p*-value was less than 0.05, the candidate gene was considered to be located in the heterozygous fragment, otherwise it was considered to be located in the homozygous fragment.

### 4.5. Genetic Map Construction and QTL Analysis

Mapmaker/Exp3.0 was used to construct the genetic linkage map [37], while the composite interval mapping method included in the WinQTLCart software version 2.5 was used to perform QTL scanning on population phenotype data and genetic maps, as well as to screen QTLs with a logarithm of odds (LOD) threshold of 2.0 [38].

## 5. Conclusions

Using back-cross populations of Cypress and KY131, a genetic linkage map with 151 polymorphic markers was constructed and 37 QTLs for grain shape, chalkiness, and amylose content were detected through QTL mapping analysis. Among them, three QTLs for grain shape (*qGW3*, *qGL7.2*, and *qLWR7.1*) and four QTLs for chalkiness (*qWBR1.2*, *qBR2*, *qWCR3*, and *qWCR11*) presented repeatable and detectable genetic effects in different generation populations. Through progeny testing, we fine-mapped *qWCR3* to a 100 kb region that contains 16 annotated genes, of which only *LOC_Os03g45210* is specifically expressed in seed and endosperm, with an 11 bp InDel variation on the second exon between two parents leading to a frameshift mutation and premature protein termination. As a result, *LOC_Os03g45210* could be the candidate gene for *qWCR3*. The results of this study lay the foundation for map-based cloning of these QTLs.

## Figures and Tables

**Figure 1 ijms-25-04389-f001:**
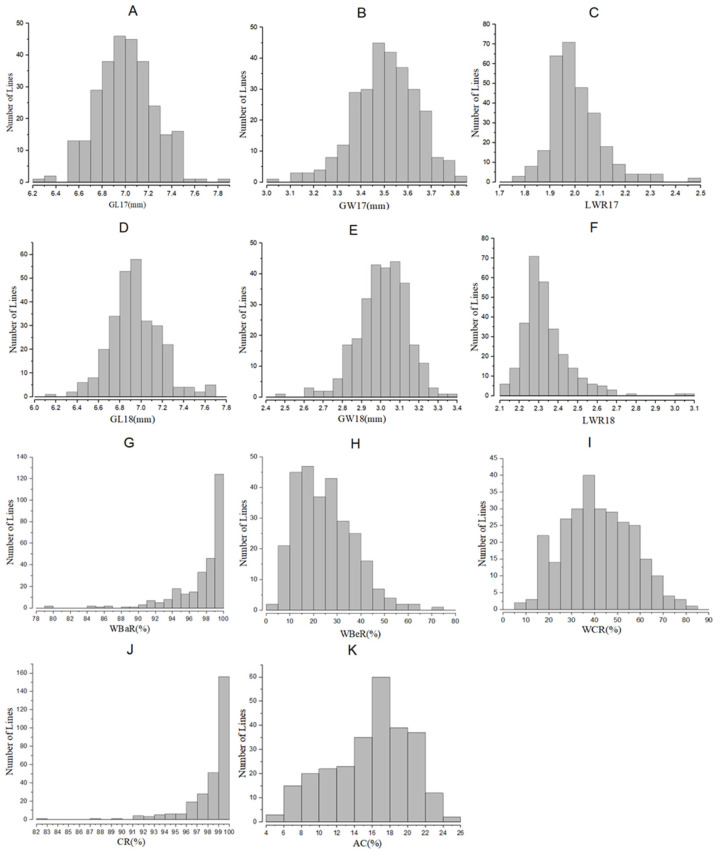
Distribution of phenotypic values among the BC_3_F_2_ and BC_3_F_2:3_ populations derived from KY131/Cypress. Grain shape-related traits in 2017 (**A**–**C**), grain shape-related traits in 2018 (**D**–**F**), chalkiness-related traits (**G**–**J**), and amylose content (**K**) in 2018. GL, grain length; GW, grain width; LWR, length-to-width ratio; WBaR, white-back rate; WBeR, white-belly rate; WCR, white-core rate; CR, chalkiness rate; AC, amylose content.

**Figure 2 ijms-25-04389-f002:**
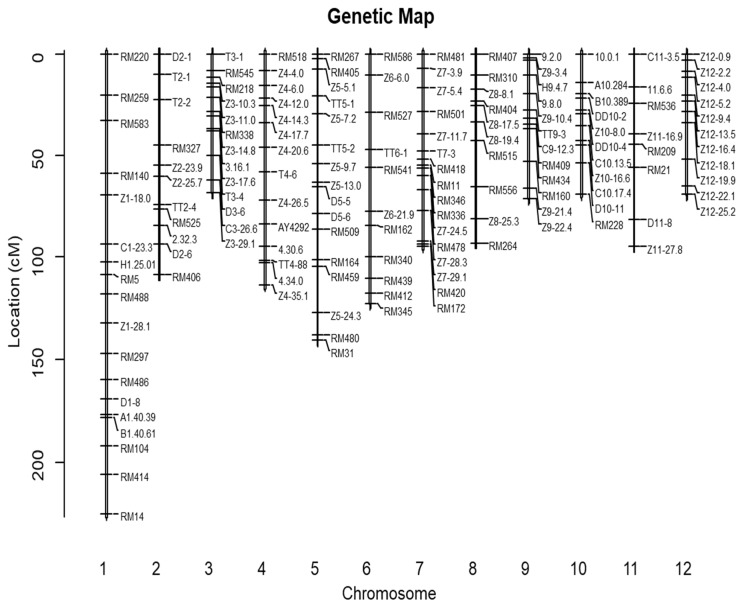
Genetic linkage map based on 151 SSR markers. RM primers were derived from public databases, while the other InDel primers were designed based on parental sequence variations.

**Figure 3 ijms-25-04389-f003:**
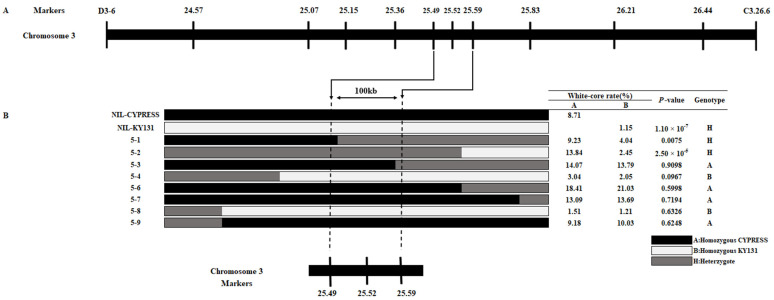
Fine mapping of *qWCR3*: (**A**) markers used in the fine mapping of *qWCR3*. (**B**) Progeny test of the eight recombinants narrowed *qWCR3* to a 100 kb region. Black, white, and grey blocks represent the genotype of homozygous CYPRESS, homozygous KY131, and heterozygote, respectively.

**Table 1 ijms-25-04389-t001:** Phenotypes of grain traits in the BC_3_F_2_ and BC_3_F_2:3_ populations.

Trait	Parents		2017HN BC_3_F_2_				2018WH BC_3_F_2:3_			
	KY131	Cypress	Mean	SD	MIN	MAX	Mean	SD	MIN	MAX
GL (mm)	6.81	8.67	6.99	0.29	5.37	7.87	6.94	0.01	6.14	7.67
GW (mm)	2.88	2.52	3.50	0.13	3.05	3.84	3.00	0.01	2.47	3.38
LWR	2.39	3.46	2.01	0.10	1.79	2.46	2.33	0.01	2.11	3.08
BR (%)	22.06	0.99					97.48	3.32	79.21	100
WCR (%)	3.82	3.16					41.38	15.08	7.66	80.72
WBR (%)	2.96	1.56					24.91	12.27	3.72	74.89
CR (%)	24.42	4.98					98.41	2.24	82.67	100
AC (%)	16.67	25.06					15.81	4.49	5.63	24.51

Note: GL, grain length; GW, grain width; LWR, length-to-width ratio; BR, white-back rate; WBR, white-belly rate; WCR, white-core rate; CR, chalkiness rate; AC, amylose content. BC_3_F_2_ was planted in Hainan (HN) in 2017. BC_3_F_2:3_ was planted in Wuhan (WH) in 2018.

**Table 2 ijms-25-04389-t002:** Correlation analysis of traits in the BC_3_F_2:3_ (KY131/Cypress) population.

	GL (mm)	GW (mm)	LWR	BR (%)	WCR (%)	WBR (%)	CR (%)
GW (mm)	0.20 **						
LWR	0.50 **	−0.74 **					
BR (%)	0.06	0.01	0.04				
WCR (%)	−0.04	0.28 **	−0.28 **	0.14 *			
WBR (%)	−0.12 *	0.24 **	−0.29 **	−0.01	0.55 **		
CR (%)	0.03	0.08	−0.04	0.93 **	0.22 **	0.13 *	
AC (%)	−0.04	−0.01	−0.02	0.03	0.24 **	0.01	0.04

Note: GL, grain length; GW, grain width; LWR, length-to-width ratio; BR, white-back rate; WBR, white-belly rate; WCR, white-core rate; CR, chalkiness rate; AC, amylose content. ** Correlation is significant at *p* < 0.01; * Correlation is significant at *p* < 0.05.

**Table 3 ijms-25-04389-t003:** QTL mapping of KY131/Cypress populations.

Traits	QTL	Chr	Interval	BC_3_F_2_			BC_3_F_2:3_		
				LOD	ADD	V (%)	LOD	ADD	V (%)
GL	*qGL3*	3	T3-1-RM545	11.46	1.291	20.60			
	*qGL4*	4	Z4-20.6-Z4-26.5	3.06	−0.188	4.85			
	*qGL6*	6	Z6-6.0-TT6-1				4.81	−0.140	5.88
	*qGL7.1*	7	RM481-Z7-5.4				3.69	0.125	4.55
	*qGL7.2*	7	T7-3-RM478	6.72	−0.342	9.12	8.74	−0.316	12.82
	*qGL7.3*	7	RM478-Z7-28.3	4.03	−0.252	7.07			
	*qGL8*	8	RM515-Z8-25.3	4.44	−0.178	6.61			
GW	*qGW1*	1	RM486-A1.40.39				2.96	0.097	4.58
	*qGW3*	3	T3-1-RM545	7.87	0.608	17.52			
	*qGW5.1*	5	RM405-TT5-1	3.49	0.156	10.03			
	*qGW5.2*	5	Z5-5.1-Z5-7.2				6.35	0.139	13.03
	*qGW7.1*	7	RM346-RM478	5.83	0.196	12.50			
	*qGW7.2*	7	T7-3-RM478				5.36	0.140	8.94
LWR	*qLWR3*	3	Z3-14.8-Z3-17.6				3.83	−0.087	4.10
	*qLWR5*	5	Z5-5.1-Z5-7.2	6.20	−0.095	8.82	10.56	−0.179	21.13
	*qLWR7.1*	7	RM346-RM478	25.80	−0.237	33.26	17.73	−0.246	26.86
	*qLWR7.2*	7	T7-3-RM346				10.19	−0.203	13.20
	*qLWR9*	9	9.8.0-Z9-10.4				3.72	−0.275	14.78
BR	*qBR1*	1	A1.40.39-RM104				2.69	6.398	4.25
	*qBR2*	2	T2-1-T2-2				4.37	2.790	8.94
	*qBR9*	9	9.2.0-9.8.0				3.03	2.680	3.79
	*qBR11*	11	C11-3.5-11-6.6				3.77	−2.135	5.05
WBR	*qWBR1.1*	1	C1-23.3-RM5				3.24	6.771	4.32
	*qWBR1.2*	1	D1-8-B1.40.61				4.52	−9.274	5.46
	*qWBR5*	5	Z5-5.1-Z5-7.2				4.54	8.925	6.73
	*qWBR6*	6	RM586-Z6-6.0				9.00	−19.770	18.17
WCR	*qWCR3*	3	D3-6-C3.26.6				2.54	−10.607	4.88
	*qWCR5.1*	5	Z5-5.1-TT5-1				2.98	10.873	5.75
	*qWCR5.2*	5	TT5-1-TT5-2				2.81	7.307	3.99
	*qWCR11*	11	C11-3.5-11.6.6				2.98	−1.310	4.47
	*qWCR12*	12	Z12-19.9-Z12-25.2				4.38	−10.997	7.02
CR	*qCR1.1*	1	Z1-18.0-C1-23.3				3.25	3.393	26.30
	*qCR1.2*	1	C1-23.3-RM5				4.04	1.448	5.59
	*qCR2*	2	T2-1-T2-2				3.43	1.675	7.22
	*qCR9*	9	H9.4.7-9.8.0				3.48	1.880	4.57
	*qCR11*	11	11-6.6-Z11-16.9				2.88	−1.026	3.76
AC	*qAC4*	4	Z4-6.0-Z4-14.3				2.52	3.687	4.01

Note: QTL, quantitative trait locus; GL, grain length; GW, grain width; LWR, length-to-width ratio; BR, white-back rate; WBR, white-belly rate; WCR, white-core rate; CR, chalkiness rate; AC, amylose content; Chr, chromosome; LOD, logarithm of odds; ADD, additive effect, where positive values indicate that alleles from Cypress increase the trait scores; V, variance explained by the QTL.

**Table 4 ijms-25-04389-t004:** Validation of QTLs in BC_3_F_3_ and BC_3_F_4_ populations.

Year	QTL	Cypress Genotype	KY131 Genotype	*p* Value	ADD
2019WH	*qLWR7.2*	2.89 ± 0.17	2.58 ± 0.20	0.0016	0.15
	*qBR2* (%)	40.05 ± 20.93	61.97 ± 18.97	0.033	−10.96
	*qWBR1.2* (%)	25.65 ± 11.52	8.66 ± 7.50	0.0019	8.50
2020WH	*qGL7.2* (mm)	8.42 ± 0.39	7.70 ± 0.24	0.0059	0.36
	*qGW3* (mm)	3.15 ± 0.15	3.31 ± 0.08	0.0069	−0.08
	*qLWR7.2*	2.80 ± 0.07	2.50 ± 0.06	5.58 × 10^−9^	0.15
2021WH	*qBR2* (%)	29.06 ± 6.19	41.81 ± 9.02	0.0026	−6.37
2021HN	*qWCR3(%)*	14.47 ± 5.90	2.46 ± 2.99	0.001	6.01
	*qWCR11(%)*	3.91 ± 1.26	14.86 ± 3.86	6.14 × 10^−7^	−5.47
2022WH	*qWCR3(%)*	8.71 ± 4.29	1.16 ± 0.75	1.09 × 10^−7^	3.78
	*qWCR11(%)*	6.08 ± 4.24	12.02 ± 6.46	0.0070	−2.97

Note: QTL, quantitative trait locus. WH, Wuhan. HN, Hainan. GL, grain length. GW, grain width. LWR, length-to-width ratio. BR, white-back rate. WBR, white-belly rate. WCR, white-core rate. ADD, additive effect; positive values indicate that alleles from Cypress increase the trait scores.

**Table 5 ijms-25-04389-t005:** The ORFs in the fine-mapping region of *qWCR3.*

Gene	Gene Product Name
*LOC_Os03g45150*	LTP family protein precursor
*LOC_Os03g45160*	Hypothetical protein
*LOC_Os03g45170*	Amino acid permease
*LOC_Os03g45180*	Expressed protein
*LOC_Os03g45194*	Oxidoreductase
*LOC_Os03g45210*	Plant Cysteine oxidase-3
*LOC_Os03g45220*	Expressed protein
*LOC_Os03g45230*	Expressed protein
*LOC_Os03g45250*	Plant Cysteine oxidase-2
*LOC_Os03g45260*	Vesicle transport v-SNARE protein
*LOC_Os03g45270*	CS domain containing protein
*LOC_Os03g45280*	Dehydrin
*LOC_Os03g45290*	Ankyrin repeat domain-containing protein 50
*LOC_Os03g45300*	Transposon protein
*LOC_Os03g45310*	Hypothetical protein
*LOC_Os03g45320*	Dehydrogenase

## Data Availability

The datasets generated during the current study are available from the corresponding author upon reasonable request.

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
