# Peer review of "QTL Analysis for Rice Quality-Related Traits and Fine Mapping of qWCR3"

_ijms, 2024, doi:10.3390/ijms25084389_

Round 1

Reviewer 1 Report (Previous Reviewer 2)

Comments and Suggestions for Authors

This is a revised version of the original manuscript IJMS02572446 reviewed in August 2023. The new re-submitted version IJMS-2919972 is improved: figure legends are modified, also the content of some paragraphs (for example, 2.4, 2.6, Conclusions) is revised.

There are, however, some notes which should be taken into account.

1.       It is necessary to edit the Abstract section.

2.       I advise to modify the Key words list: they should not duplicate the title.

3.       In the original comments I recommended to extend section 4. Materials and Methods. However, I have the following questions to the new version:

-          by what means the parameters white core rate (WCR), white belly rate (WBR), white back rate (BR) and chalkiness rate (CR) have been evaluated? The appropriate information is absent in the section "Materials and Methods".

-          I recommend to include references to the methods of DNA extraction used in the work (Lines 285-287). There are many modifications of CTAB method, moreover, the TPS method is a recent modification, and is still not widely used.

Author Response

For research article

Response to Reviewer 1 Comments

1. Summary

2. Questions for General Evaluation

Reviewer’s Evaluation

Response and Revisions

Does the introduction provide sufficient background and include all relevant references?

Yes

Are all the cited references relevant to the research?

Yes

Is the research design appropriate?

Yes

Are the methods adequately described?

Can be improved

We have requested professional experts to revise the entire text.

Are the results clearly presented?

Yes

Are the conclusions supported by the results?

Yes

3. Point-by-point response to Comments and Suggestions for Authors

Comments 1: [It is necessary to edit the Abstract section.]

Response 1: Thank you for pointing this out. We agree with this comment. Therefore, we have made modifications to the abstract section.

Comments 2: [I advise to modify the Key words list: they should not duplicate the title.]

Response 2: Thank you for your suggestion. We have modified it.

Comments 3: [By what means the parameters white core rate (WCR), white belly rate (WBR), white back rate (BR) and chalkiness rate (CR) have been evaluated? The appropriate information is absent in the section "Materials and Methods".]

Response 3: We apologize for our negligence. We have made modifications to this section (Line 292-296).

Comments 4: [I recommend to include references to the methods of DNA extraction used in the work (Lines 285-287). There are many modifications of CTAB method, moreover, the TPS method is a recent modification, and is still not widely used.]

Response 4: We apologize for our negligence. We have added the corresponding reference (Line 305 & 441).

Reviewer 2 Report (New Reviewer)

Comments and Suggestions for Authors

The manuscript titled "QTL Analysis for Rice Quality-related Traits and Fine Mapping of qWCR3" used two populations of BC3F2 and BC3F2:3 from two jopanica rice KY131 and Cypress to identified QTLs for rice grain shape, chalkiness and amylose content. They have detected 37 QTLs and fine mapped one gene for qWCR3. The topic of this study is significance and interesting. Here are some comments for this manuscript.

1. line 12, "several quality related traits" should be deleted.

2. line 12-14, "In this study, a genetic map with 151 polymorphic markers was construct QTL (Quantitative Trait Locus) mapping for rice grain shape, chalkiness and amylose content were performed." is confusing. Please rewrite this sentence.

3. line 19, "LOC_Os45210" should be LOC_Os03g45210.

4. line 71, more information for the BC3F2:3 population is needed in the material section.

5. line 83, is BC3F3 right? is it BC3F2:3?

6. the qWCR3 was mapped to the gene LOC_Os03g45310. However, no genetic evidence has been provided. The author might need to constructed a knockout mutant for this gene to support thier results.

Comments on the Quality of English Language

Language editing is needed.

Author Response

Reviewer 3 Report (New Reviewer)

Comments and Suggestions for Authors

The manuscript entitled "QTL Analysis for Rice Quality-related Traits and Fine Mapping of qWCR3" offers an insightful investigation into the genetic elements affecting rice quality, with a particular emphasis on grain shape, chalkiness, and amylose content, all of which are vital for the marketability of rice. The research provides notable revelations regarding the genetic underpinnings of rice quality attributes. The precise mapping of qWCR3 within a 100kb region and the identification of potential candidate genes, such as LOC_Os03g45210 which displays sequence variation leading to a frameshift mutation, deserve mention. This identifies a specific genetic modification that could be harnessed in breeding programmes aimed at mitigating chalkiness, an undesirable characteristic in rice quality. The creation of an elaborate genetic linkage map utilising 151 polymorphic markers is praiseworthy. Such detailed mapping allows for the more accurate pinpointing of QTLs linked to essential quality traits in rice, demonstrating a systematic approach to unravelling complex genetic traits. The study adeptly correlates phenotypic variations with genetic markers, thereby enriching our understanding of the impact of specific genetic regions on rice quality traits. This cohesive analysis is essential for the application of genetic discoveries in practical breeding strategies. The objectives of this study are clearly articulated, and the findings are of interest. While I have no significant critiques, I would like to offer some suggestions for consideration below:

Identifying candidate genes within the qWCR3 region marks a considerable advancement. Nevertheless, the research could gain from functional validation experiments to definitively ascertain the roles of these genes in chalkiness regulation. For instance, gene knockout or overexpression studies could conclusively verify the function of each candidate gene.

The research mainly concentrates on the genetic determinants of rice quality, without thoroughly investigating how these genetic factors might interact with environmental conditions to affect trait manifestation. Grasping gene-environment interactions is vital for developing rice varieties that are adaptable to a range of climatic conditions.

Although the research successfully pinpoints specific candidate genes, it might benefit from a broader examination of the gene networks and pathways in which these genes are involved. Understanding the interaction between LOC_Os03g45210 and other genes implicated in endosperm development and stress responses could offer a more comprehensive perspective on the genetic regulation of rice quality.

The study could be enriched by contemplating epigenetic changes that could influence the expression of pivotal genes within the qWCR3 region. Epigenetic factors may significantly impact trait inheritance and expression, particularly in response to environmental stresses.

While the research provides genetic insights, a more detailed exposition on the biochemical mechanisms by which the identified genetic variations influence rice quality traits would be valuable. Elucidating how the frameshift mutation in LOC_Os03g45210 disrupts the protein's function and its consequent effects on cellular processes related to chalkiness could deepen the mechanistic understanding.

The above suggestions could be considered for a more in-depth discussion.

Comments on the Quality of English Language

Regarding the English writing, the manuscript is generally lucid and succinct, effectively transmitting complex scientific concepts and findings without undue wordiness. This clarity is essential for readers to comprehend the study's aims, methods, results, and implications. The employment of specific technical vocabulary and terminology is fitting for the subject matter, reflecting a high level of expertise in genetics and plant science. This specialised language accurately conveys the study's details and nuances. While the writing is clear, the incorporation of a broader variety of sentence structures could improve the text's readability and dynamism. For instance, blending complex sentences with simpler ones could render the text more engaging. The use of transitional phrases between sections and paragraphs could be enhanced to bolster the narrative's coherence. Effective transitions can guide the reader through the argument or discussion, underscoring the links between various points. A thorough proofreading session could identify and amend any minor grammatical or typographical errors, thereby enhancing the overall professionalism of the document.

Author Response

This manuscript is a resubmission of an earlier submission. The following is a list of the peer review reports and author responses from that submission.

Round 1

Reviewer 1 Report

Comments and Suggestions for Authors

The manuscript "QTL mapping and validating for grain quality in Rice" presents the mapping of Qtl for grain shape, chalkiness and amylose content. The authors found 37 Qtl in two Back-cross populations, they also fine-mapped one of the Qtl (qWCR3) finding two candidate genes.

Comments on the Quality of English Language

The manuscript needs some improvement in the english, minor changes.

Author Response

Response to reviewer #1
The manuscript needs some improvement in the English, minor changes.

Response: Thank you for your kindly guidance, we will correct the language as much as possible.

Reviewer 2 Report

Comments and Suggestions for Authors

The research aims at the genetic mapping and validation of genetic factors determining variation of several important characters of grain quality, i.e., grain shape characteristics, chalkiness and amylase content. The topic of the study is original. To date, several major genes responsible for these traits are known but other factors constituting the genetic architecture of the traits are still unknown. The QTLs were mapped to 12 rice chromosomes with the use of 151 SSR-markers developed in the study. Six of them have been successfully validated.

Without doubt, the authors performed a large volume of experimental work. However, the presentation of the material is superficial in some places, the manuscript contains stylistic and grammatic and should be improved. I recommend to extend section 4 Materials and Methods and to describe experimental procedures in more details. This will facilitate a better understanding of the results.

The conclusions are generally consistent with the evidence and arguments presented. However, the statement “LOC_Os03g45210 showed a significantly different expression level in endosperm between Cypress and KY131 (line 290-291) is not supported by any experimental data. Paragraph 2.6 contains information on the sequence difference between the two parents and the information on the special expression of this gene only (lines 172-175).

The list of cited literature contains 36 bibliographic references. All references are appropriate.

The manuscript contains five tables and three figures. All figures have good quality, but the legends could be extended.

 Additional comments and recommendations:

  Please edit the title. A variant: QTL Mapping and Validating for Grain Quality Traits in Rice

-          The abstract is too brief. Please extend

-          Line 10 – Please edit the sentence “Rice quality is the main factor that determines the market competitiveness, which is coregulated by multiple genes”. A variant: “Rice quality is the main factor that determines the market competitiveness. The seed quality traits in rice are coregulated by multiple genes”.

-          Line 55 – Please use in plural: “rice cultivars”

-          Lines 55-56 – Incorrect sentence: “In this study, a high-generation backcross population derived from two rice cultivar (Cypress and KY131) was developed and used for QTL mapping of rice grain shape, chalkiness, and AC”. Please edit. A variant: “In this study, a backcross population derived from two rice cultivar (Cypress and KY131) was developed and used for QTL mapping of several traits, grain shape, chalkiness, and AC”. Two and three generations can not be considered as high generations number

-          Line 58 – Please replace “was detected” by “were detected”

-          Lines 60-61 – Please complete the sentence: “By using progeny test method, qWCR3 was fine mapped to a 100 kb region of chromosome 3

-          Line 72 - Correlation analysis of all traits in BC3F2:3 population showed a highly significant positive correlation between GL and LWR

-          Line 74 – Please replace “correlated” by “correlation”

-          Line 78 – Please replace “population” by “populations”

-          Line 79 – Please replace “length width rate” by “length to width rate”

-          Line 84 – Please edit the name of table 2 “Correlation analysis of BC3F2:3 population of KY131/Cypress”. A variant: “Correlation analysis of traits in the BC3F2:3 (KY131/Cypress) population”

-          Please pay attention to the evaluation of correlation coefficients. Some of then look very weak (not significant).

-          Line 90 – Please replace “population” by “populations”

-          Line 99 – Please replace “molecular” by “SSR”

-          Line 128 – Please replace “length width” by “length to width”

-          Line 134 – Please replace “seed” by “seeds”

-          Lines 139-140 Please edit the sentence: “The additive effects of all QTLs showed similar to those of QTL mapping”.

-          Line 142 – Please replace “population” by “populations”

-          Line 169 – Please replace “exist” by “exhibit”

-          Line 184 – Please replace “clone” by “cloning”

-          Lines 278-279 – Please replace “polyacrylamide gels” by “polyacrylamide gel electrophoresis

-          What does it mean R2 %? In table 3? Please specify.

Author Response

Response to reviewer #2
(1) Without doubt, the authors performed a large volume of experimental work. However, the presentation of the material is superficial in some places, the manuscript contains stylistic and grammatic and should be improved. I recommend to extend section 4 Materials and Methods and to describe experimental procedures in more details. This will facilitate a better understanding of the results.

Response: Thank you for your suggestion. We have modified the section 4 Materials and Methods.

(2) The conclusions are generally consistent with the evidence and arguments presented. However, the statement “LOC_Os03g45210 showed a significantly different expression level in endosperm between Cypress and KY131 (line 290-291) is not supported by any experimental data. Paragraph 2.6 contains information on the sequence difference between the two parents and the information on the special expression of this gene only (lines 172-175).

Response: Sorry for our mistake. We have corrected it in the section 5 Conclusions, please check it.

(3) The manuscript contains five tables and three figures. All figures have good quality, but the legends could be extended.

Response: Thank you for your suggestion. We have added the legends to Figures, please check it.

(4) Please edit the title. A variant: QTL Mapping and Validating for Grain Quality Traits in Rice

Response: Thank you for your suggestion. The title of this manuscript has been changed to “QTL Analysis for Rice Quality-related Traits and Fine Mapping of qWCR3”, please check it.

(5) The abstract is too brief. Please extend

Response: Thank you for your suggestion. We have modified it.

(6) Line 10 – Please edit the sentence “Rice quality is the main factor that determines the market competitiveness, which is coregulated by multiple genes”. A variant: “Rice quality is the main factor that determines the market competitiveness. The seed quality traits in rice are coregulated by multiple genes”.

Response: Thank you for your suggestion. We have modified it.

(7) Line 55 – Please use in plural: “rice cultivars”

Response: Sorry for our mistake. We have corrected it, please check.

(8) Lines 55-56 – Incorrect sentence: “In this study, a high-generation backcross population derived from two rice cultivar (Cypress and KY131) was developed and used for QTL mapping of rice grain shape, chalkiness, and AC”. Please edit. A variant: “In this study, a backcross population derived from two rice cultivar (Cypress and KY131) was developed and used for QTL mapping of several traits, grain shape, chalkiness, and AC”. Two and three generations can not be considered as high generations number

Response: Thank you for your kindly suggestion. Incorrect sentence “In this study, a high-generation backcross population derived from two rice cultivar (Cypress and KY131) was developed and used for QTL mapping of rice grain shape, chalkiness, and AC” has been corrected to “In this study, two backcross populations derived from two rice cultivars (Cypress and KY131) was developed and used for QTL mapping of rice grain shape, chalkiness, and AC”, please check it.

(9) Line 58 – Please replace “was detected” by “were detected”

Response: Sorry for our mistake. We have corrected it, please check.

(10) Lines 60-61 – Please complete the sentence: “By using progeny test method, qWCR3 was fine mapped to a 100 kb region of chromosome 3

Response: Sorry for our mistake. We have corrected it, please check.

(11) Line 72 - Correlation analysis of all traits in BC3F2:3 population showed a highly significant positive correlation between GL and LWR

Response: Sorry for our mistake. We have changed it to “Correlation analysis was conducted for all traits in BC3F2:3 population. Among grain shape related traits, a extremely significant positive correlation was detected between GL and LWR”, please check.

(12) Line 74 – Please replace “correlated” by “correlation”

Response: Sorry for our mistake. We have corrected it, please check.

(13) Line 78 – Please replace “population” by “populations”

Response: Sorry for our mistake. We have corrected it, please check.

(14) Line 79 – Please replace “length width rate” by “length to width rate”

Response: Sorry for our mistake. We have corrected it, please check.

(15) Line 84 – Please edit the name of table 2 “Correlation analysis of BC3F2:3 population of KY131/Cypress”. A variant: “Correlation analysis of traits in the BC3F2:3 (KY131/Cypress) population”

Response: Thank you for your kindly suggestion. We have corrected it, please check.

(16) Please pay attention to the evaluation of correlation coefficients. Some of them look very weak (not significant).

Response: Thank you for your kindly suggestion. We have modified the description of section 2.1 Phenotype variation in BC3F2 and BC3F2:3 populations, please check it.

(17) Line 90 – Please replace “population” by “populations”

Response: Sorry for our mistake. We have corrected it, please check.

(18) Line 99 – Please replace “molecular” by “SSR”

Response: Sorry for our mistake. We have corrected it, please check.

(19) Line 128 – Please replace “length width” by “length to width”

Response: Sorry for our mistake. We have corrected it, please check.

(20) Line 134 – Please replace “seed” by “seeds”

Response: Sorry for our mistake. We have corrected it, please check.

(21) Lines 139-140 Please edit the sentence: “The additive effects of all QTLs showed similar to those of QTL mapping”.

Response: Sorry for our mistake. We have deleted this mistake description, please check it.

(22) Line 142 – Please replace “population” by “populations”

Response: Sorry for our mistake. We have corrected it, please check.

(23) Line 169 – Please replace “exist” by “exhibit”

Response: Sorry for our mistake. We have corrected it, please check.

(24) Line 184 – Please replace “clone” by “cloning”

Response: Sorry for our mistake. We have corrected it, please check.

(25) Lines 278-279 – Please replace “polyacrylamide gels” by “polyacrylamide gel electrophoresis”

Response: Sorry for our mistake. We have corrected it, please check.

(26) What does it mean R2 %? In table 3? Please specify.

Response: Sorry for our mistake. We have corrected the “R2(%)” to “V(%)” , please check it.

We would like to take this opportunity to thank you for all your time involved and this great opportunity for us to improve the manuscript. We hope you will find this revised version satisfactory.

Sincerely,

The Authors